# MXene/Ferrite Magnetic Nanocomposites for Electrochemical Supercapacitor Applications

**DOI:** 10.3390/mi13101792

**Published:** 2022-10-20

**Authors:** Arun Thirumurugan, Ananthakumar Ramadoss, Shanmuga Sundar Dhanabalan, Sathish-Kumar Kamaraj, Natarajan Chidhambaram, Suyambrakasam Gobalakrishnan, Carolina Venegas Abarzúa, Yerko Alejandro Reyes Caamaño, Rednam Udayabhaskar, Mauricio J. Morel

**Affiliations:** 1Sede Vallenar, Universidad de Atacama, Costanera #105, Vallenar 1612178, Chile; 2Advanced Research School for Technology & Product Simulation (ARSTPS), School for Advanced Research in Polymers (SARP), Central Institute of Petrochemicals Engineering & Technology (CIPET), T.V.K. Industrial Estate, Guindy, Chennai 600032, Tamil Nadu, India; 3Functional Materials and Microsystems Research Group, RMIT University, Melbourne, VIC 3000, Australia; 4Tecnológico Nacional de México, Instituto Tecnológico El Llano, El Llano 20330, Mexico; 5Department of Physics, Rajah Serfoji Government College (Autonomous), Bharathidasan University, Thanjavur 613005, Tamil Nadu, India; 6Department of Nanotechnology, Noorul Islam Centre for Higher Education, Deemed to be University, Kumaracoil 629180, Tamil Nadu, India; 7Instituto de Investigaciónes Científicasy Tecnológicas (IDICTEC), Universidad de Atacama, Copayapu 485, Copiapo 1531772, Chile

**Keywords:** MXene, ferrites, supercapacitor applications

## Abstract

MXene has been identified as a new emerging material for various applications including energy storage, electronics, and bio-related due to its wider physicochemical characteristics. Further the formation of hybrid composites of MXene with other materials makes them interesting to utilize in multifunctional applications. The selection of magnetic nanomaterials for the formation of nanocomposite with MXene would be interesting for the utilization of magnetic characteristics along with MXene. However, the selection of the magnetic nanomaterials is important, as the magnetic characteristics of the ferrites vary with the stoichiometric composition of metal ions, particle shape and size. The selection of the electrolyte is also important for electrochemical energy storage applications, as the electrolyte could influence the electrochemical performance. Further, the external magnetic field also could influence the electrochemical performance. This review briefly discusses the synthesis method of MXene, and ferrite magnetic nanoparticles and their composite formation. We also discussed the recent progress made on the MXene/ferrite nanocomposite for potential applications in electrochemical supercapacitor applications. The possibility of magnetic field-assisted supercapacitor applications with electrolyte and electrode materials are discussed.

## 1. Introduction

MXene, a class of 2D transition metal carbides/nitrides, has been identified as a new emerging material for various applications including energy storage, electronics, and bio-related due to its wider physical characteristics. Further, the formation of hybrid composites of MXene with other materials introduces interesting uses in multifunctional applications. MXene offers tremendous promise for energy storage applications, due to its lamellar construction, outstanding density, conductivity, configurable terminations, and ionic pseudocapacitance charge storage process [1,2,3]. A wide variety of MXenes are produced by diverse mixtures of transition metals, carbon and/or nitrogen, and diverse n layers; however, only a few have been generated and employed in energy storage applications thus far, with Ti_3_C_2_Tx being the most investigated [4,5,6,7,8,9]. The possible MAX phase associated with MXene, compositions, and structure of typical MXene and some of the experimentally prepared MXenes are shown in Figure 1. By removing the A layers from the MAX phase, MXenes are created. The compound is recognized as MXene due to its composition, which is M_n+1_AX_n_, where M is an initial transition metal, A is primarily an element from group IIIA or IVA (i.e., 13 or 14), X is C and/or N, and n is either 1, 2, or 3. The composition of the MAX phase and MXene compound, tuning of composition of MXene and the possible MXene could be understood from Figure 1a–c. MXenes represent a new large family that extends the world of 2D materials, and there has been an increase in interest in 2D materials other than graphene. Distinctive morphologies, strong electronic conductivities, and diverse chemistries, among other distinctive features of MXenes, have made them desirable in numerous applications, particularly for energy storage [10]. Due to its renowned etching processes and the extensive theoretical and experimental investigations on its physicochemical and electrochemical characteristics, Ti_3_C_2_T_x_, one prominent example among the MXene group, is a key interest of many. The electrochemical characteristics of MXene have been debated in recent years with some experimental evidence [3,6,11,12,13]. In aqueous electrolytes, hydrated cations are intercalated into MXene electrodes. Within the constrained potential window, the hydrated cations establish an electric double layer throughout the interlayer region to provide a standard capacitance. When nonaqueous electrolytes are utilized, the limited solvation shell must progressively breakdown due to the significant internal potential difference in the interlayer, even when solvated ions are diffused into the interlayer during the early phase of charging. Desolvated ions only intercalate when charged more, and a donor band is formed when the desolvated cations’ atomic orbitals cross over with those of MXene. By the charge transition from the metal ions to MXene, the reduction of MXene caused by the donor band development results in an intercalation pseudocapacitance. The detailed charge storage mechanism was explored by a few recent studies [3,13,14,15]. Further the improvement in the charge storage capacity could be enhanced through the surface modification process [16].

Spinel ferrites are homogeneous substances that have the typical chemical form of AB_2_O_4_, where A and B are metallic ions that are located at 2 distinct crystallographic places, tetrahedral (A sites) and octahedral (B sites), respectively, with key elements of Fe^3+^ in their structure. Three distinct spinel ferrite configurations, termed as normal, inverse, and mixed systems, are attainable for the composition of MFe_2_O_4_, based on the position of M^2+^ and Fe^3+^ location selection. M^2+^ is distributed at the tetrahedral location and Fe^3+^ at the octahedral location in a typical spinel configuration of ferrite. Fe^3+^ is evenly located at both positions in an inverse spinel configuration, whereas M^2+^ solely resides at the octahedral location. Both ions are indiscriminately located at the tetrahedral and octahedral positions in a mixed spinel configuration. Fe_3_O_4_ is considered an intriguing material for energy storage systems due to additional benefits such as high theoretical-specific capacitance (2299 F/g), a wide potential window (−1.2 to 0.25 V), abundance in nature, relative inexpensiveness, lesser toxicity, etc., [17]. The charge storage mechanism in various ferrites was thoroughly analyzed by various experimental methods in supercapacitor applications [17,18,19,20]. All these kinds of ferrite could be easily prepared by selecting suitable precursors for suitable experimental procedures. 

The MXene/Ferrite composites have attracted much attention in various applications including batteries, electrochemical supercapacitors, photocatalytic degradation, electromagnetic shield, and water purification methods [23,24,25,26,27,28,29]. The charge storing capacity of various MXenes and ferrites was not achieved towards its theoretical capacity. There are many ongoing investigations focused on improvement in the electrochemical performance of different MXenes, as well as on the various ferrites by adjusting the shape, size, and functional groups. The formation hybrid materials could be advantageous to achieve better outcomes from both materials due to the synergic effect. 

The advantages of these magnetic nanocomposites over other available nanocomposites are its magnetic characteristics. These magnetic nanocomposites can be reused after the completion of any specific application. The collection processes of the magnetic nanocomposites are also easy to process with the external magnet because of the magnetic nature of the nanocomposites. Another advantage in this magnetic nanocomposite is the utilization of the eternal magnetic field during the application process. The presence of the external magnetic field with the magnetic nanocomposite definitely could influence the reaction mechanism, which could reflect in the efficiency of the nanocomposites in specific applications [30]. Green synthesis methods are also possible in the formation of these kinds of composites to obtain environmental sustainability. The effect of the external magnetic field on a few ferrites and non-magnetic materials has been studied. However, the optimization of external field strength for various magnetic and non-magnetic materials is much needed. The composite of MXene with ferrites requires attention as there are many opportunities to explore more nanocompositions. 

## 2. Synthesis Methods

### 2.1. Preparation of MXene

MXene can be prepared through a top-down approach whereby the etching takes place using bottom-up methods, including chemical vapor deposition, pulsed laser deposition etc. [31,32,33]. However, the bottom-up approach has some limitations including the availability of precursors and desired size, which could be a challenging process through the chemical synthesis or chemical exfoliation process. The etching, delamination and filtration processes comprise the basic preparation process for scalable synthesis in the MAX phase [34]. Through a simple process that involves selectively eliminating the A component from the MAX form at room temperature, MXene synthesis can be performed [35]. In this procedure, the MAX phase substance is agitated in aqueous HF at a predetermined fraction for a predetermined amount of time, separated from the blends by centrifugation and/or filtering, and then washed by deionized water until the pH of the dispersion attains values between 4–6. The overall process is schematically shown in Figure 2.

The etching process can be done through hydrogen fluoride (HF), lithium fluoride (LiF)/HF, NH_4_HF_2_, molten salt etching, alkali treatment and the intercalation of additives such as hydrazine with N,N-dimethylformamide and dimethyl sulfoxide, and urea, tetrabutylammonium hydroxide, tetramethylammonium hydroxide, amine n-butylamine, and choline hydroxide [36,37,38,39,40,41,42,43,44,45,46]. As all these synthesis processes are progressing rigorously towards a better synthesis process, there are numerous opportunities to solve the present issues related to the scalable single-phase synthesis of MXene. Studies have etched Ti_3_C_2_ MXene and Ti_2_C MXene with LiF, potassium fluoride (KF), ammonium fluoride (NH_4_F), and sodium fluoride (NaF) in hydrochloric acid (HCl), and then delaminated the resultant material with urea, dimethylsulfoxide, or ammonium hydroxide. Further optimizations were done with the time and temperature variation of the reaction process. The observed morphology of Ti_3_C_2_ and Ti_2_C for various conditions is shown in Figure 3.

HF, which is extremely corrosive and detrimental to the functioning of Li-ion batteries and supercapacitors, is obviously used in the majority of the MXenes production processes. The preparation of a typical MXene—Ti_3_C_2_T_x_ (T = OH, O) using an alkali-assisted hydrothermal technique has recently been reported for fluorine free synthesis of MXene [44]. This technique was modeled after a Bayer procedure that is frequently employed in the refinement of bauxite. Multilayer Ti_3_C_2_T_x_ with a purity of 92 weight percent was produced by the complete procedure, which uses 27.5 M NaOH at 270 °C. Two stages make up the OH^−^ attack on the Al layers in Ti_3_AlC_2_: oxidizing the Al to form Al (oxide) hydroxides and dissolving the Al (oxide) hydroxides in alkali as shown in Figure 4. Ti_3_AlC_2_’s exterior Al atoms are susceptible to oxidation and dissolve in NaOH as Al hydroxide is soluble in NaOH. These results in -OH or -O terminate the exposed Ti and Al atoms. The inner Al ions also oxidize if this etching procedure is allowed to continue. Al(OH)_3_ and its AlO(OH) are unavoidably produced by the OH^−^ attack, though. These insoluble compounds are prevented from easily interacting with OH to form soluble Al(OH)_4_^−^ due to the lattice confining provided by the Ti layers (Figure 4a). This kind of “jamming effect” must be removed because it prevents the synthesis of MXene. The Bayer procedure demonstrates that dissolving the aluminum (oxide) hydroxides requires considerable temperatures and larger alkali concentrations. This is ideal for getting rid of the obstructing Al compounds and will allow the entire selective etching process to move in the direction of formation of MXenes. The overall process is schematically shown in Figure 4. The SEM, TEM and HRTEM micrograph of prepared MXenes are shown in Figure 4b–d.

The stability of MXene is one of the important parameters to consider in any type of application [48,49]. In general, the MXene is easily oxidized or degraded from its original physicochemical characteristics. The oxidation of the Ti_3_C_2_ results in the formation of Ti-based oxide materials on the surface or anywhere in the MXene along with amorphous carbon. However, several procedures were demonstrated to control the oxidation or degradation process of MXenes [50]. A few of the processes include the synthesis of MXene with fewer defects, surface modification of MXene with organic ligands or any other suitable surface modifier, dispersions of MXenes in organic solutions, and the storing of MXene in solution, which is stored inside the inert gas-filled container [49,51]. If the MXene hybrids with polymeric materials or any other stable materials are required in particular applications, then the degradation of the stability and electronic conductivity of MXene could be reduced due to the polymeric or any other stable materials in hybrids [52].

### 2.2. Synthesis of Ferrites

Numerous methods are available for the synthesis of various ferrites including co-precipitation, sol-gel, hydrothermal, solvothermal, the polyol process, and the chemical oxidation process [53,54,55,56,57,58,59]. However, the selection of the synthesis process is important to achieve the required shape and size of the desired ferrite. Chemical methods are preferred for the preparation of various ferrites due to the opportunity to tune the ferrite particle shape and size by modifying the experimental parameters including precursors, additives, reaction time, reaction temperature, and post-thermal treatment [60,61]. The co-precipitation method is a very common chemical method used to synthesize all kinds of ferrites. The co-precipitation method is a simple reaction process, and it can be completed within one hour depending on the ferrite stoichiometry.

The basic reaction mechanism for the formation of ferrites through the co-precipitation process is,
(1)$$M^{2+}+M^{3+}+NaOH\;\to MFe_{2}O_{4}+NaCl+H_{2}O$$

The divalent metal ions can be varied to form various forms of ferrites and doping of other metal ions, also a simple process in this reaction. Mono-sized ferrite particle formation is possible with optimized experimental parameters.

The chemical oxidation process is also a widely used chemical synthesis process for the formation of ferrites with larger size, and this method could be used to prepare the ferrites with a range of particle sizes. However, the formation of mono sized particles is challenging, and a wide range of particle size distribution is usually observed. The ferrite particle size can be tuned by varying the oxidant, precipitant and also by introducing the nucleating agent. 

The general reaction process may be defined as,
(2)$$M^{2+}+KNO_{3}+NaOH\;\to MFe_{2}O_{4}+KNO_{2}+NaCl+H_{2}O$$

Several other methods were utilized for the development of various ferrites with tunable size, variable morphology by varying the experimental parameters, addition to the surfactant and doping process. Through the thermal decomposition process with the help of oleic acid (OA) and oleylamine (OAm), monodisperse magnetic Mn-Zn ferrite with a variety of morphologies from nanocrystals to clusters such as spherical, cubical and starlike nanocrystal have been effectively produced [62]. The reaction temperature and aging period affect whether nanocrystals will form. The fluctuation of the monomer fraction in the traditional La Mer model serves to assess the crystal nucleation/growth process in synthesis. For samples taken from all synthesis below 260 °C, no substance is seen. When the monomer is consistently utilized, irregular “crystal nuclei” (2–3 nm) are created at 260 °C, as shown in Figure 5A. At a later temperature of 260–300 °C, the first smaller-sized spheroidal nanocrystals are formed. When the aging period is increased from 20 to 40 min at 300 °C, a large number of uniform spherical nanocrystals in a closed packed array are generated. Small quasi-spherical shapes are initially produced in cubical and starlike morphology formation at the earlier temperatures (260–280 °C), as observed in Figure 5B and C. The result is mostly spherical with a tiny number of quasi-cubical shapes at the reaction’s initial stage at 300 °C. Most of the nanocrystals remain growing as the aging duration at 300 °C is increased from 20 to 40 min, generating uniform cubical morphology with changes in size as shown in Figure 5B or continuing to form the eight corners of the half-formed precursors to create starlike nanocrystals as observed in Figure 5C. By altering the OA/OAm ratios, it is possible to successfully witness the uninterrupted shape transformation of nanocrystals from their initial sphericity, cubicity, to their final starriness, as schematically shown in Figure 5D.

Using a quick and affordable thermal decomposition approach, it was possible to successfully produce spherical, cubical, and hexagonal shaped Co-ferrite nanoparticles [63]. The TEM micrograph and particle size distribution of the developed cubical, spherical, and hexagonal shaped particles are shown in Figure 6. The optical band gaps, which vary according to the shape of the particles, were set between 1.41 and 1.32 eV. At 10 K and room temperature, respectively, hexagonal nanoparticles showed a superior coercivity of 17,845.8 and 1577.76 Oe than spherical and cubical shaped particles. Because hexagonal-shaped Co-ferrite has an extraordinary surface-to-volume ratio and supplementary structural facets, the coercivity can be altered. Due to the morphology-dependent cation allocation, different shaped particles have adjustable saturation magnetization, which is important for certain room temperature magnetic applications. The observed magnetic characteristics along with the calculated magnetic anisotropy values are listed in Table 1.

Without the use of a surfactant, a single moderate hydrothermal technique at 160 °C was used to synthesize Ni-ferrite with the morphologies of spheres, rods, and nano-octahedrons [64]. When the pH level was 7, nano-spheres with sizes of 10–25 nm were produced. At a pH level of 12, nano-rods with a length of around 1 m and width of 50–60 nm were generated. With a pH of 13, the sample’s octahedral shape was discovered. Each side is approximately 150 nm long. Clearly, the Ni-ferrite nano-octahedrons’ surface contains a few barely detectable nanoparticles. The schematic process for the formation mechanism of various shapes is shown in Figure 7. The saturation magnetization of nanooctahedrons and nanorods was measured as 55 and 40 emu/g. The spherical shaped particles did not show any saturation magnetization as the particle sizes are in the superparamagnetic regime. This synthesis strategies could be adopted for the preparation of ferrite nanoparticles with different morphology where it may be required for the specific applications.

Doping of metal ion into ferrite also could be used to alter the size and shape of the ferrite [65,66,67,68,69,70]. It is also noted that the particle size and shape influences the magnetic characteristics of the particle [71,72,73,74]. For the preparation of mixed ferrite, the divalent and trivalent metal ions can be introduced and adjusted for the complete phase formation with required cation distribution [75,76]. Similarly, other chemical methods such as the sonochemical process, solid state reaction, and auto-combustion method also could be adopted for the synthesis of various ferrite magnetic nanoparticles with the required shape and size [77,78,79,80].

### 2.3. MXene/Ferrite Composite Formation

The formation of composite is important as the application may require a synergic effect from hybrid materials. It is advantageous for the MXene to not have any nanomaterials between the layers to avoid the restacking. If the nanocomposite formation process is not completely done, then there is a chance of the formation of two different materials without any physical or chemical attachment. To achieve the complete formation of composites of ferrites and MXenes, first the MXene should be covered with the ferric, ferrous, or any other metal ions before the formation of ferrites. The further addition of oxidant or reducing agent, depending on the experimental methods, could form the ferrites physically, chemically or physic-chemically attached on the surface of MXenes. Any of the above discussed chemical synthesis processes could be adopted for the formation of MXene composites with MXene. Further, the addition of any other attractive materials such as carbon-based materials is also possible for the formation of ternary composites with MXene.

The attachment of ferrites on MXene is important to be confirmed from any characterization techniques. The presence of ferrites on MXene could be evidenced from X-ray diffraction studies. But the manner of ferrite formation on the MXene could be evidenced through any microscopic methods. The fraction of ferrites in the MXene composites could be estimated from the magnetic hysteresis measurement, by estimating the saturation magnetization of bare ferrite and MXene/ferrite composites. The vibration modes can be analyzed through Fourier transform infrared spectroscopy and Raman spectroscopy methods. Further, the evidence of the metallic state of the metal ions present in the MXene composites could be analyzed from the X-ray photoelectron spectroscopy method. The specific surface area, pore size, and volume could be measured from the BET surface area measurement techniques. If we have strong evidence for the physicochemical characteristics of the MXene/ferrite, then the nanocomposite could be utilized for electro chemical energy storage applications.

Solvothermal synthesis was adopted for the synthesis of MXene/Fe_3_O_4_ nanocomposite with uniform distribution of Fe_3_O_4_ on the exterior and between the interlayers of MXene [81]. The simple schematic synthesis process was showed in Figure 8a. The particle sizes with 50–100 nm were observed for the Fe_3_O_4_ nanoparticles on the surface of MXenes in the MXene/Fe_3_O_4_ nanocomposites. Similarly, ultrasonic treatment is used to synthesize the MXene/Fe_3_O_4_ nanocomposite for the potential applications as additive sulfur cathode in the lithium−sulfur battery [29]. The schematic synthesis process for the Fe_3_O_4_ distributed on MXene nanocomposite formation is shown in Figure 8. The even distributions of Fe_3_O_4_ on the lamellar structure of Mxene were achieved and the improvement in the battery performance was evidenced. 

The successful nanocomposite formation was achieved by Xiaojun Yang et al. [27] with MXene and Fe_3_O_4_ through the hydrothermal, chemical etching, and vacuum filtration technique. The MXene nanosheets with the thickness of around 1 nm were decorated with the spherical nanoparticles of Fe_3_O_4_ with a size of a few dozen nanometers. The thin layer structure of MXene and the attachment of spherical Fe_3_O_4_ nanoparticles on the MXene layers were clearly evidenced from the TEM micrograph as shown in Figure 9.

Various forms of ferrites with MXene along with other materials have been developed and their physicochemical characteristics studied for various applications. Table 2 summarizes a few available combinations of MXene nanocomposites and their applications.

From Table 2, it is clearly understood that there is much information available on the preparation of MXene/ferrites, which were utilized for the photocatalytic degradation of dyes, water purification and electromagnetic wave absorption. Few reports are available for the application of MXene/ferrites in batteries and fuel cells. However, very limited reports are available on the electrochemical super capacitive characteristics of MXene/ferrites nanocomposites. So, it is important to study the physicochemical and electrochemical characteristics of various MXene/ferrite nanocomposites to be utilized in next generation energy storage systems.

## 3. MXene/Ferrite Electrode for Supercapacitor Applications

The utilization of multivalence metal ions in the mixed ferrite could be beneficial for the charge transfer process during electrochemical reactions. In this regard, Co-ferrite (CoFe_2_O_4_) has been attempted for the formation of nanocomposites with MXene due to the extraordinary theoretical specific capacitance, mixed valence, and chemical stability [24]. Co-ferrite is an established example of an inverse spinel in which Fe^3+^ ions coexist with Co^2+^ ions at tetrahedral regions and Fe^3+^ ions at octahedral regions, resulting in enhanced redox activity and effective charge storage. The main issue with the Co-ferrite in supercapacitor applications is its low electronic conductivity. However, this could be solved by forming hybrid composites with other materials that have excellent electronic conductivity. The addition of Co-ferrite would be advantageous in MXene/Co-ferrite nanocomposites, as the Co-ferrite could be used to avoid the restacking of MXene layers. Co-precipitation with wet chemical etching processes were utilized for the formation of Co-ferrite and MXene. The MXene/Co-ferrite composite formation was achieved through the ultra-sonication process for one hour and dried at 80 °C in vacuum. The single phase of Co-Ferrite was evidenced for bare Co-ferrite nanoparticles as well as for the MXene/Co-ferrite nanocomposite, and the observed XRD pattern of bare Co-ferrite and MXene/Co-ferrite are shown in Figure 10i(a,b). The characteristics reflections of MXene were not witnessed in the XRD pattern of MXene/Co-ferrite nanocomposite. However, the presence of MXene in the MXene/Co-ferrite nanocomposite was discussed through the observation if peak broadening occurs, variation in the lattice parameters and shift in the characteristic peak is observed. The observed peak broadening and peak shift are shown in Figure 10i(c,d). The SEM micrograph confirms the Co-ferrite nanoparticles sizes ranging from 70–90 nm and the layers construction of MXene as shown in Figure 10ii(a–c). The uniform dispersion of Co-ferrite over the MXene layered structure was evidenced from the SEM micrograph as shown in Figure 10ii(d). It is also noted that the Co-ferrite nanoparticles were also incorporated between MXene layers, which would be advantageous for the supercapacitor applications.

The average layer thickness was estimated as 1 nm from atomic force microscopic experiments. The electrochemical supercapacitor analysis of the individual and nanocomposites were assessed in 1 M KOH electrolyte in a potential window of 0.0 to 0.5 V. The cyclic voltammogram curve shows the two distinct anodic and cathodic points, which might be due to the transition of Fe^2+^/Fe^3+^ and Co^2+^/Co^3+^ during the electrochemical redox reactions. The overall electrochemical reactions with the MXene/Co-ferrite nanocomposite were explained thorough the following reactions,
(3)$$CoFe_{2}O_{4}+H_{2}O+OH^{-}\to 2\;FeOOH+CoOOH+e^{-}$$
(4)$$CoOOH+OH^{-}\;\to CoO_{2}+H_{2}O+e^{-}$$
(5)$$FeOOH+H_{2}O\;\to Fe{\left(OH\right)}_{3}\;\to {\left(FeO_{4}\right)}^{2-}+3e^{-}$$

The better electrochemical super capacitive performance was observed for MXene/Co-ferrite compared with bare Co-ferrite and MXene. The detailed electrochemical studies were carried out for the MXene/Co-ferrite nanocomposites. The CV curves were measured at the scan rate of 10 to 100 mV/s and the measured CV curves are shown in Figure 11a. The charge/discharge curves were measured at the current density of 1 to 8 A/g and the observed curves are shown in Figure 11b. The specific capacitance was estimated from the Galvanostatic charge-discharge (GCD) curves and calculated Cs values corresponding to its current densities are shown in Figure 11c. The highest Cs value of 1268 F/g was achieved at a current density of 1 A/g, whereas, with the same current density, bare Co-ferrite and MXene showed a specific capacitance value of 594 and 1046 F/g, respectively. The long-term cyclic stability was evidenced up to 5000 cycles with a current density of 7 A/g, and the stability curve is shown in Figure 11d. The calculated specific capacity corresponding to its current densities is shown in Figure 11e. The highest specific capacity value of 440 C/g at a current density of 1 A/g was achieved with the lower charging transfer resistance values of 0.25 Ω.

The observed electrochemical super capacitive characteristics of MXene/Co-ferrite indicate that these kinds of composites could be further investigated for the improvement in the specific capacitance and cyclic stability.

A recent study shows that the selection of electrolyte is also important for the MXene based ferrite composites [92]. The bare Fe_3_O_4_ and MXene/ Fe_3_O_4_ magnetic nanocomposites were developed through the modified chemical oxidation process by optimized experimental conditions. A few µm-sized MXenes and particle sizes ranging from 80 to 160 nm of Fe_3_O_4_ were evidenced from the microscopic measurements. The attachment of Fe_3_O_4_ on the MXene sheets in the MXene/Fe_3_O_4_ nanocomposite was evidenced from the TEM micrograph as shown in Figure 12.

Three electrolytes of LiCl, KOH and Na_2_SO_4_ were used to evaluate the suitable electrolyte for MXene/Ferrite composites with reduced graphene oxide. The measured CV, GCD curve are shown in Figure 13a,b. The higher area was observed for the nanocomposites characterized in LiCl electrolyte. Also, the LiCl showed better charge/discharge characteristics for MXene/Fe_3_O_4_ nanocomposite, compared with other two electrolytes. The estimated Cs with various current densities is shown in Figure 13c along with the EIS spectra in Figure 13d. The observed electrochemical performance demonstrated that the LiCl could provide better performance, which might be due a better combination of the lowest Li ion with the higher pore sizes in the MXene/ferrite composites. However, this electrolyte needs to be evaluated for other various combinations of MXene/ferrites to generalize the electrolyte characteristics. 

## 4. Magnetic Field-assisted Applications

The external magnetic field can influence the electrochemical reactions, which is evidenced for different external magnetic fields in various electrolytes, electrodeposition of materials, and electrochemical applications [132,133,134,135,136,137,138]. It is expected that the magnetic field will be built as an effective system with the lowest energy consumption for large-scale applications through the continual and systematic development of this field [139,140,141,142]. Numerous investigations have demonstrated that the electron’s energy state could be improved under a magnetic field. As a consequence, the improved energy state of the electrons would undoubtedly enhance the capacitance efficiency by increasing the effectiveness of electron movement at the electrolyte-electrode interface.

The experimental setup for the magnetic field-assisted supercapacitor applications can be easily constructed. The external magnetic field could be produced by two electromagnets, and the strength of the field could be controlled by adjusting the current supply to the electromagnets. The schematic experimental setup for the evaluation of role of the external magnetic field is shown in Figure 14. In the magnetic field produced by two electromagnets, a three-electrode cell should be immersed. To maintain consistent magnetic field intensity distribution between the positive and negative electrodes, the cell must be precisely placed. To assess the impact of the magnetic field, five different types of aqueous electrolytes, including NaOH, KOH, NaCl, K_2_SO_4_, and H_2_SO_4_ were tested [143]. On the working electrode, an amorphous commercial AC structure with significant microporosity was utilized. 

The various impacts of the magnetic field on the magnetocapacitance factor (MCF) values for NaOH, KOH, H_2_SO_4_, NaCl, and K_2_SO_4_ electrolytes with a magnetic field of 876 and 1786 Oe are summarized in Figure 15A,B. One may hypothesize that the active ions for rendering MCF are most likely proton and hydroxide ions by comparing H_2_SO_4_ and KOH with K_2_SO_4_ electrolytes (or by comparing NaOH with NaCl in the situation of the similar cation or anion). Without adding a magnetic field, the initial conductivities of the KOH, NaOH, and H_2_SO_4_ electrolyte were significantly higher than those of NaCl and K_2_SO_4_ electrolytes, as observed in Figure 15C. The KOH, NaOH, and H_2_SO_4_ electrolyte’s conductivities linearly increased as the magnetic field’s strength was amplified. Additionally, when the magnetic field was applied, the viscosity of all acidic and alkaline electrolytes decreased as observed in Figure 15D, which also contributes to the increase in conductivity. In contrast, in the magnetic field, the conductivities of K_2_SO_4_ and NaCl are barely changed. This may be because the neutral electrolyte does not clearly show the polarization impact of metal cations or anions in water. 

The impact of magnetic fields on the electrochemical behavior of KOH electrolytes at various concentrations was studied as shown in Figure 16A,B and compared the MCF values in KOH electrolytes of various concentrations at scan rates of 10 and 200 mV/s. The MCF values generally increased with concentration (6 M) at the slow scan rate of 10 mV/s, and the rise degree was more pronounced in the presence of a higher magnetic field. The MCF values, however, were nearly zero for concentrations of 8 and 10 M, independent of how strong the magnetic field was. When the concentrations were 1, 2, 4, and 6 M, however, the MCF values were positive at the relatively high rate of 200 mV/s. The MCF values, however, monotonously decreased as the concentration rose at a higher fraction of KOH (8 and 10 M). The original conductivity steadily increased with the rise in the fraction of KOH up to 8 M, as illustrated in Figure 16C. The conductivity decreased as the concentration was raised to 10 M, which is connected to the ions’ delayed migration because of a high chance of ion collision and powerful electrostatic contact between anions and cations. The initial viscosity rose monotonously with the higher concentration, as seen in Figure 16D. When the electrolyte fraction was 1, 2, and 4 M, the magnetic field reduced the viscosity; however, in 6, 8, and 10 M KOH electrolytes, the viscosity rose. Concentrated electrolytes’ decreased conductivity and increased viscosity in a magnetic field would hinder ion transport, which would affect the MCF readings. The limiting current density (I_lim_) values derived from these Tafel polarization curves are displayed in Figure 16E. When compared to the initial outcomes obtained without adding the magnetic field, I_lim_ enlarged little in the 1 and 2 M KOH electrolytes and significantly in the 4 M KOH electrolyte. I_lim_ values, however, were lower than the initial values in electrolytes with 8 and 10 M KOH. The data above showed a relationship between I_lim_, the concentration of electrolytes, and the strength of the magnetic field. The values of the diffusion coefficients for the various concentrated KOH electrolytes in various magnetic fields are shown in Figure 16F.

From the discussion, it is understood that the various magnetic forces acting on the electroactive ions cause solution convection at the electrolyte interface, which is the source of the magnetic field induced capacitance variation throughout the electrochemical reaction process. The fight among the driving force brought on by the Lorentz force and the damping force brought on by the conductivity of the electrolyte in KOH electrolytes is a key factor influencing the OH transport with the magnetic field. Whereas in the H_2_SO_4_ electrolyte, in addition to the previously mentioned forces, the third paramagnetic force produced by the paramagnetic H^+^ under the magnetic field prevents the movement of H^+^. 

In a one-pot synthesis, Fe_3_O_4_/rGO nanocomposites were created by coupling the reduction of graphene oxide with the formation of Fe_3_O_4_ NPs under ambient conditions and utilized for a supercapacitor application with an external magnetic field to evaluate the influence of the external magnetic field [144]. Fe_3_O_4_ and Fe_3_O_4_/rGO hybrid symmetric two electrode electrochemical performance was examined utilizing CV and GCD measurements with an electrolyte of 1 M Na_2_SO_4_. With 0.125 T of the external magnetic field, the electrochemical performances of both electrodes have been explored (0.125 T). 

Figure 17a,b shows the CV curves of Fe_3_O_4_ and Fe_3_O_4_/RGO in a magnetic field. These curves are substantially wider than those of the electrodes without a magnetic field, which were recorded at a scan rate of 5 mV/s, and show a greater specific capacitance. Figure 17c,d show the GCD curves of Fe_3_O_4_ and Fe_3_O_4_/RGO electrodes in the presence and absence of a magnetic field, respectively, at a current density of 0.1 A/g. Calculated Cs for ferrite and ferrite/rGO electrodes in a magnetic field are 303 and 869 F/g, respectively, at a scan rate of 5 mV/s. These values are 1.45 and 1.93 times greater than those of electrodes without using the field. These outcomes could be explained by enhanced electron transfer in the Fe^2+^/Fe^3+^ redox process in the materials with magnetic field exposure, which enhances the performance of devices by improving interface charge density, facilitating electrolyte transportation, and promoting cation intercalation/de-intercalation performance.

The thermal decomposition approach was used to create consistently sized Fe_2_O_3_ nanoparticles with sizes between 12 and 30 nm for the graphene surface [145]. Decorating of the non-conductive Fe_2_O_3_, the capacitance performance of the Fe_2_O_3_/graphene nanocomposite is significantly decreased because of the higher internal resistance and poor electron transportation efficiency. The electrochemical supercapacitive characteristics were analyzed with 1.0 M Na_2_SO_4_ and the external magnetic field of 720 Gauss. The measured CV curves for bare Fe_2_O_3_ and Fe_2_O_3_/graphene nanocomposites with the external magnetic field are shown in Figure 18a–d. 

The Cs of graphene is considerably increased by 67 and 27% at 2 and 10 mV/s, respectively, after positioning the electrode in an external magnetic field. At 2 and 10 mV/s, the Fe_2_O_3_/graphene nanocomposite displays an enhancement that is much greater by 154.6 and 98.2%. The estimated energy density and current density for bare Fe_2_O_3_ and Fe_2_O_3_/graphene nanocomposites are shown in Figure 19a,b. Equivalent circuit modeling of EIS shows a significant amount of leakage resistance between the electrode-electrolyte line caused by a relaxation progression at low frequency. By applying a small magnetic field, this relaxation process was considerably restrained, effectively increasing the capacitor output. These findings provide a novel method for improving the electrochemical capacitors now in use by merely introducing an external magnetic field. We could utilize the specific response from any magnetic or nonmagnetic materials, which could be utilized to influence various applications of nanomaterials. The influence of the external magnetic field on various applications is listed in Table 3.

From the above discussions, we observed that the external magnetic field can influence the electrochemical performance of the electrolyte. On the other hand, when we use the magnetic nanomaterials as supercapacitive electrode materials, the supercapacitive performance is influenced by the external magnetic field, as the magnetic electrode materials could easily respond to the applied magnetic field. The external magnetic field may increase or decrease the electrochemical performance of the supercapacitive electrode materials subject to the magnetic nature of the materials. Also, the amount of magnetic field used as external magnetic field can influence the electrochemical supercapacitive characteristics of the magnetic electrode materials. This discussion does not conclude that the rise in field strength can increase the supercapacitive performance. The supercapacitive performance of the magnetic materials depends on the particle size, shape, magnetic characteristics of the materials, and the external magnetic field strength. Limited reports are only available on the evaluation of the magnetic field effect on the supercapacitive performance of the magnetic materials. The combinations of MXene with ferrites should be investigated further with various forms of MXenes, ferrites, and further investigation could be done under the external magnetic field.

## 5. Summary, Scope, and Future Perspective

The synthesis process for the MXene, Ferrites and MXene/Ferrite magnetic nanocomposites was briefly discussed. The importance of attachment of the ferrite on the layers of MXenes was studied. The recent progress on MXene/ferrite formation through chemical reaction was reviewed with Fe_3_O_4_ and CoFe_2_O_4_. The importance of the selection of a suitable electrolyte was analyzed with the result of MXene/Fe_3_O_4_ magnetic nanocomposites. The external magnetic field plays an important role in the tuning of electrochemical characteristics of the electrolyte, as well as electrode materials. The role of the external magnetic field on the electrolytes was discussed with KOH, NaOH, NaCl, K_2_SO_4_, and H_2_SO_4_ electrolytes with 876 and 1786 Oe of magnetic field. Furthermore, we provided a detailed analysis of the effect of the amount of KOH electrolyte with the external magnetic field. The schematic preparation process and magnetic field-assisted electrochemical supercapacitor application is shown in Figure 20.

There are lots of issues related to the synthesis of scalable and single phase MXene through top down as well from the bottom approach methods. Present research could make the challenges into opportunities to explore further on the synthesis of MXene without any impurity or secondary phase. Several strategies are also demonstrated to have a stable MXene by avoiding oxidation/degradation. However, the formation of single phase, stoichiometry variation through synthesis process, shape and size of ferrites can be varied depending on the requirement as the well-established procedures and discussions were available for various ferrites. There are numerous ferrites with different stoichiometry need to be explored for the formation of magnetic nanocomposites with MXene. Several chemical reaction processes also to be optimized for the better formation of nanocomposite with required physicochemical interaction. Green synthesis methods could also be adopted for the development of magnetic hybrid composites with MXene. By optimizing the experimental parameters such as starting precursors, reaction time, reaction temperature and post thermal treatment we could be able to achieve the perfect nanocomposites of MXene and ferrite without any agglomeration. The effect of magnetic field on the electrochemical characterization of MXene/Ferrite nanocomposite are open to be explore further for the next generation supercapacitor applications as there are more opportunities to tune the magnetic materials shape, size, magnetic characteristics, and the strength of magnetic field. Analyses of studies that have already been published, hint at this field of study’s potential significance. However, there are still many unexplored facets of the magnetic field influence on electrochemical applications which needs attention. 

## Figures and Tables

**Figure 1 micromachines-13-01792-f001:**
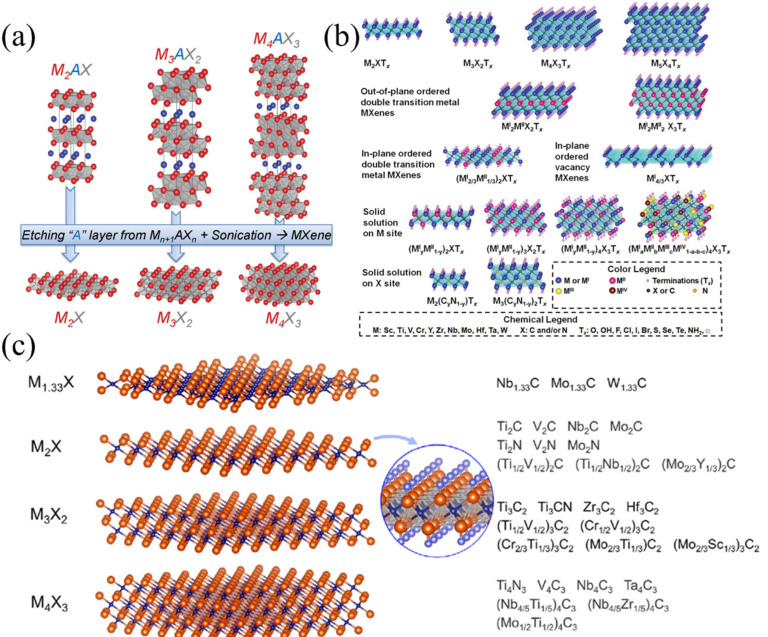
(**a**) MAX phase structures and their associated MXenes. Reproduced with permission from [5] Copyright (2013) John Wiley & Sons. (**b**) Usual MXene compositions and structures. Reproduced with permission from [21] Copyright (2021) John Wiley & Sons. (**c**) Experimentally discovered MXenes: structural and molecular formula. Reproduced with permission from [22] Copyright (2019) Elsevier.

**Figure 2 micromachines-13-01792-f002:**
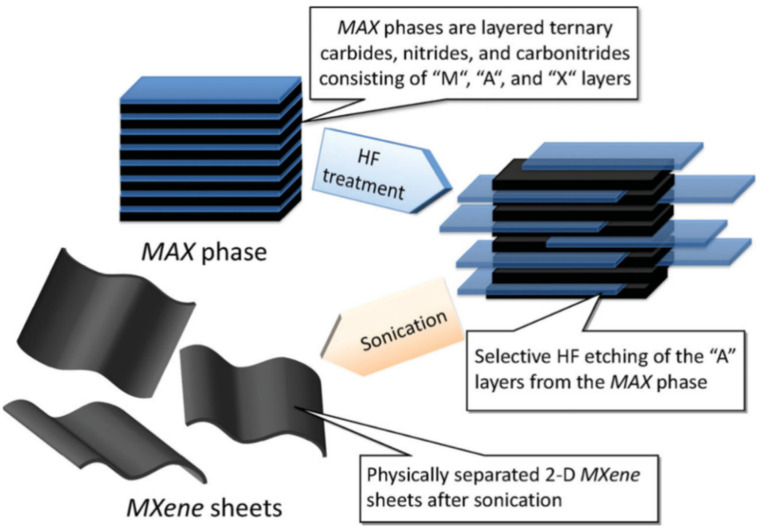
Schematic process of etching and exfoliation process of MXene from MAS phase. Reproduced with permission from [35] Copyright (2012) American Chemical Society.

**Figure 3 micromachines-13-01792-f003:**
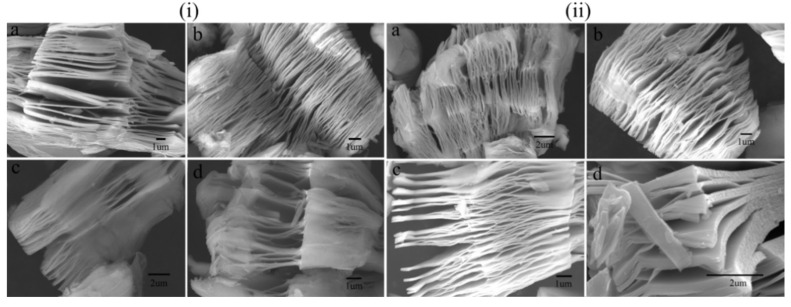
SEM micrograph of exfoliated MXene (**i**) (Ti_3_C_2_) with (**a**) LiF (**b**)NaF, (**c**) KF (**d**) NH_4_F, in HCl, and (**ii**) (Ti_2_C) with (**a**) LiF (**b**)NaF, (**c**) KF (**d**) NH4F, in HCl. Reproduced with permission from [47] Copyright (2017) Elsevier.

**Figure 4 micromachines-13-01792-f004:**
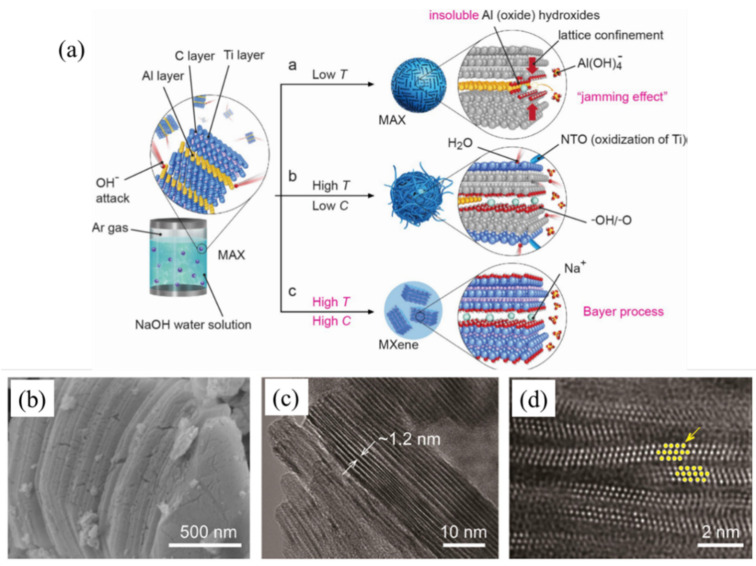
Diagram illustrating the interaction between a NaOH solution and Ti_3_AlC_2_ under various circumstances. ((**a**) At low temperatures, Al (oxide) hydroxides obstruct the extraction of Al. (**b**) In the presence of high temperatures and low NaOH conc, certain Al (oxide) hydroxides dissolve in NaOH. (**c**) Based on the Bayer mechanism, dissolving the Al (oxide) hydroxides in NaOH is aided by high temperatures and higher concentrations of NaOH), (**b**–**d**) SEM, TEM, and HRTEM micrograph of prepared MXene. Reproduced (modified) with permission from [44] Copyright (2018) John Wiley & Sons.

**Figure 5 micromachines-13-01792-f005:**
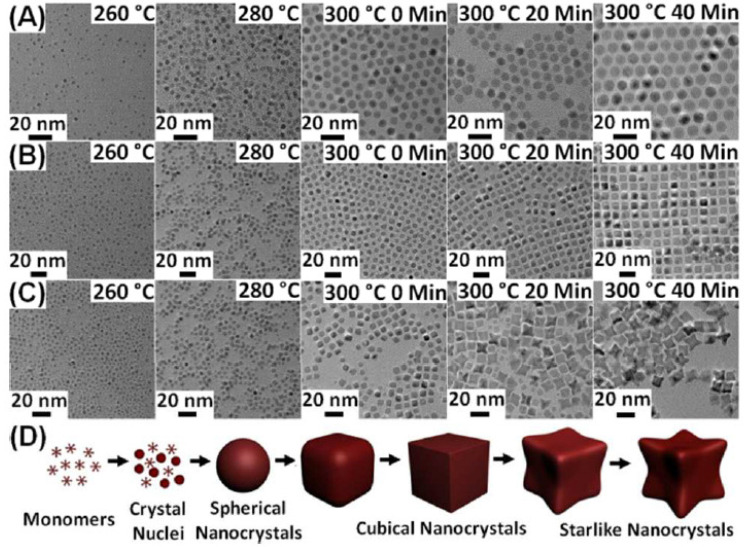
TEM micrograph of Mn-Zn ferrite nanocrystals collected from the reaction mixture at 260 °C, 280 °C, and 300 °C following aging for 0, 20, and 40 min. (**A**) spherical (**B**) cubical, and (**C**) starlike. (**D**) Evolution of the shape of starlike nanocrystals shown schematically. Reproduced with permission from [62] Copyright (2013) American Chemical Society.

**Figure 6 micromachines-13-01792-f006:**
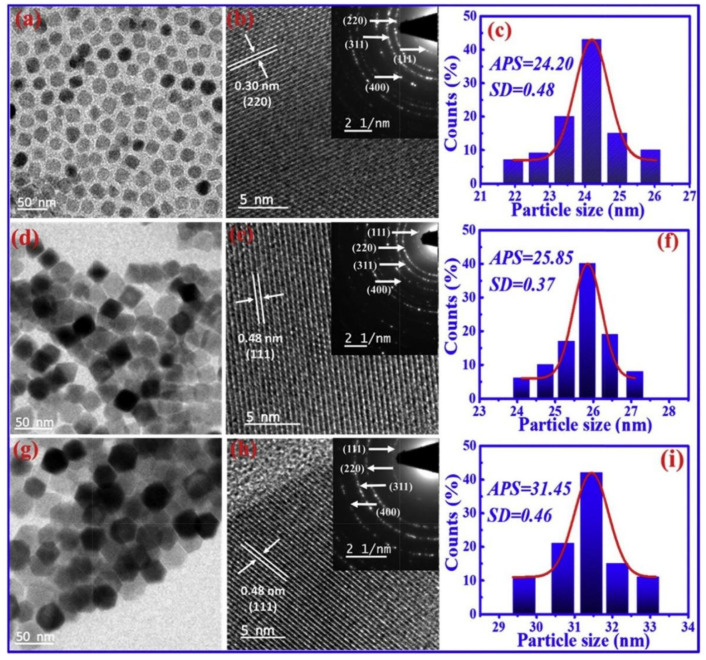
TEM, HRTEM micrograph and size distribution of (**a**–**c**) sphere-CoFe_2_O_4_ (**d**–**f**) cubic-CoFe_2_O, and (**g**–**i**) hexagonal-CoFe_2_O. Reproduced with permission from [63] Copyright (2019) Elsevier.

**Figure 7 micromachines-13-01792-f007:**
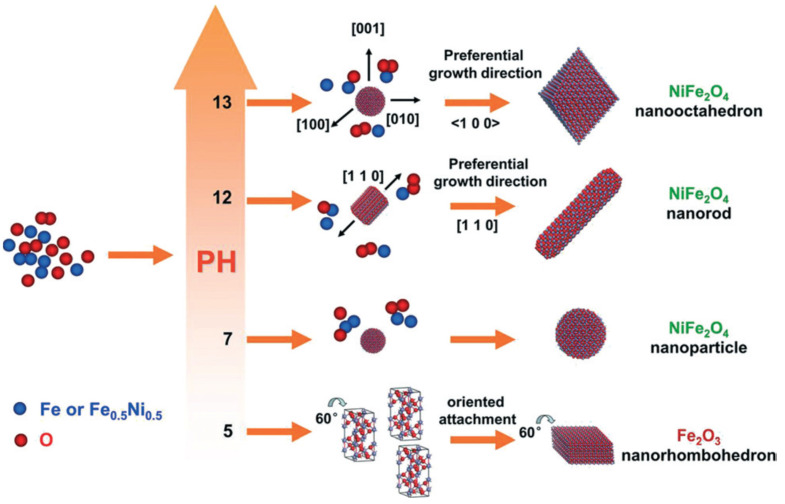
Schematic diagram for the different morphology formation processes of α-Fe_2_O_3_ and Ni-ferrite. Reproduced with permission from [64] Copyright (2015) Royal Society of Chemistry.

**Figure 8 micromachines-13-01792-f008:**
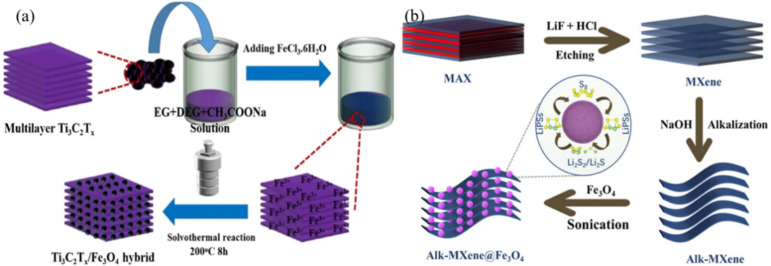
(**a**) Schematic synthesis process of Mxene/Fe_3_O_4_ nanocomposite through Solvothermal process. Reproduced with permission from [81]. (**b**) Schematic synthesis process of Alk-MXene composites with Fe_3_O_4_ through ultrasonic treatment. Reproduced with permission from [29] copyright (2022) John Wiley and Sons.

**Figure 9 micromachines-13-01792-f009:**
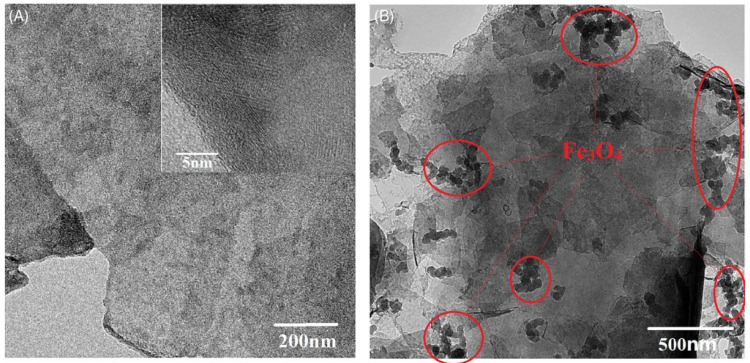
TEM micrographs of (**A**) MXene and (**B**) MXene/Fe_3_O_4_ nanocomposite. Reproduced with permission from [27] Copyright (2022) John Wiley and Sons.

**Figure 10 micromachines-13-01792-f010:**
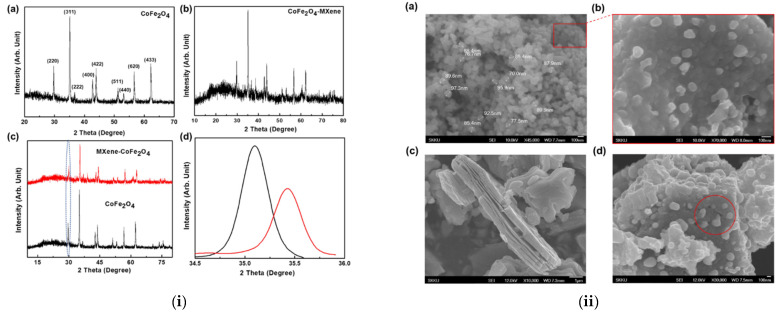
(**i**) XRD pattern of (**a**) Co ferrite, (**b**) MXene/Co-ferrite, (**c**) comparison pattern, and (**d**) magnified at lower angle, and (**ii**) SEM micrograph of (**a**,**b**) Co-ferrite, (**c**) MXene and (**d**) MXene/Co-ferrite nanocomposite. Reproduced with permission from [24] Copyright (2020) American Chemical Society.

**Figure 11 micromachines-13-01792-f011:**
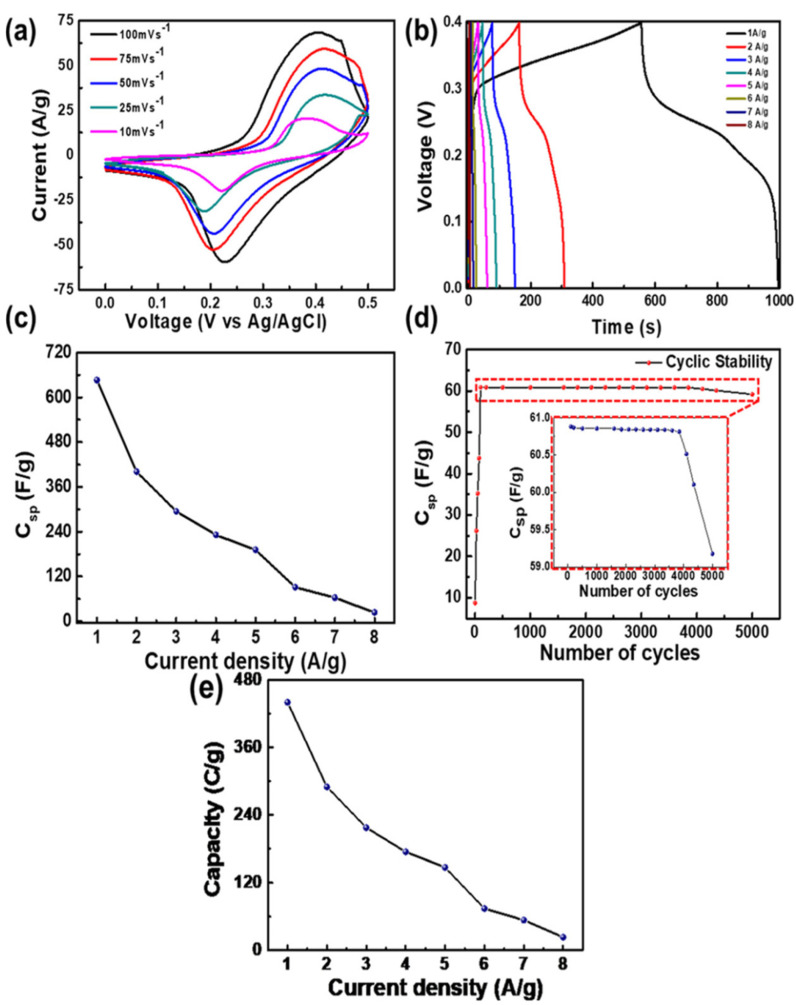
(**a**) CV, (**b**) GCD curves, (**c**) Cs vs. current density, (**d**) cyclic stability and (**e**) specific capacity vs. current density. Reproduced with permission from [24] Copyright (2020) American Chemical Society.

**Figure 12 micromachines-13-01792-f012:**
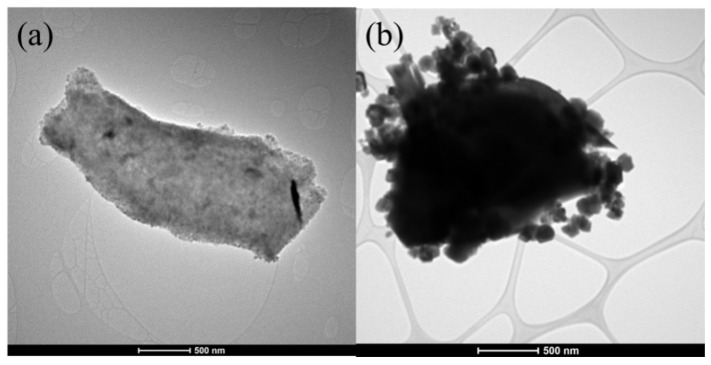
TEM Micrograph of (**a**) MXene, and (**b**) MXene/ Fe_3_O_4._

**Figure 13 micromachines-13-01792-f013:**
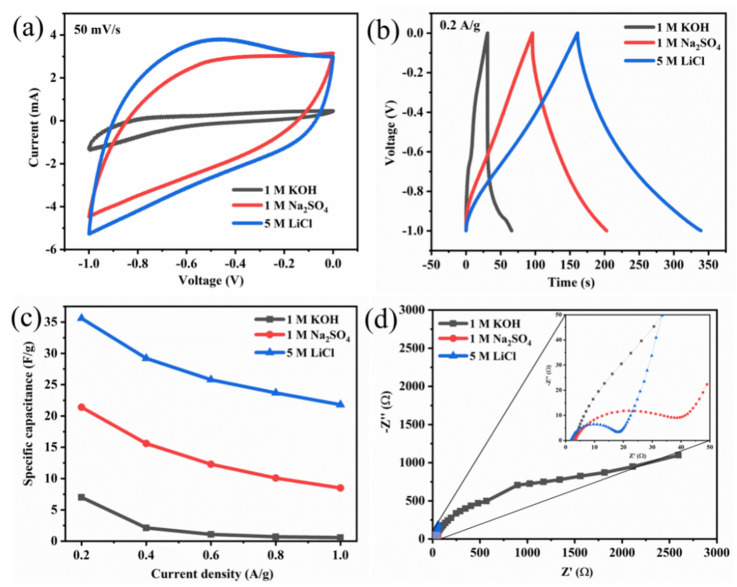
(**a**) CV, (**b**) GCD, (**c**) Cs vs. current density, and (**d**) EIS curves of MXene/Fe_3_O_4_.

**Figure 14 micromachines-13-01792-f014:**
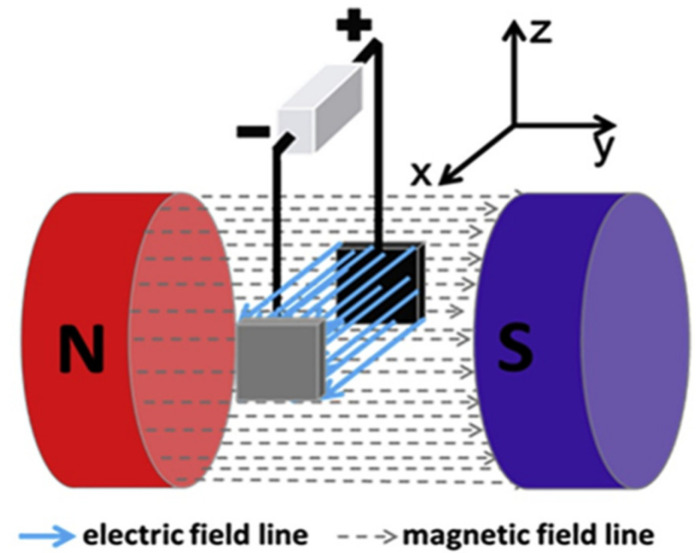
Schematic setup for the measurement of effect of external magnetic field. Reproduced (modified) from [143] Copyright (2021) with permission from Elsevier.

**Figure 15 micromachines-13-01792-f015:**
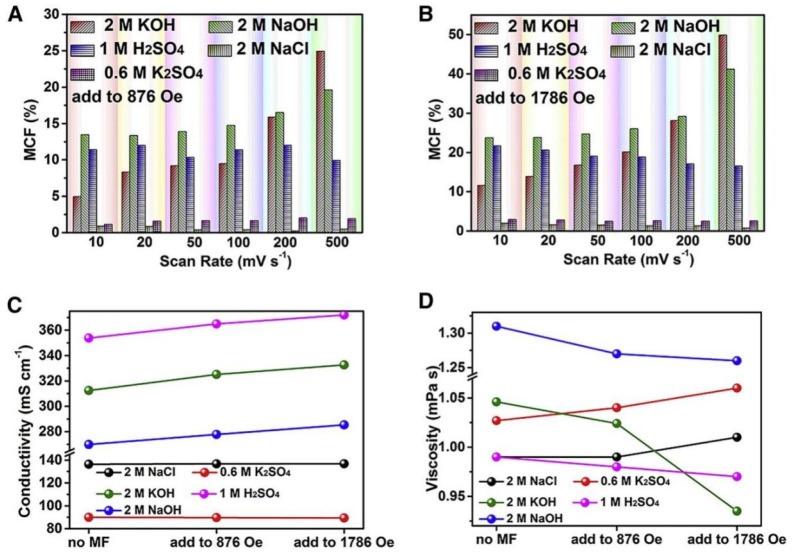
The impact of magnetic field on various aqueous electrolyte’s conductivity, viscosity, and electrochemical properties. The MCF variation with scan rates under the magnetic field of (**A**) 876 Oe, and (**B**) 1786 Oe, (**C**) Variation in the conductivity with magnetic field, and (**D**)Variation in the viscosity with magnetic field. Reproduced (modified) from [143] Copyright (2021) with permission from Elsevier.

**Figure 16 micromachines-13-01792-f016:**
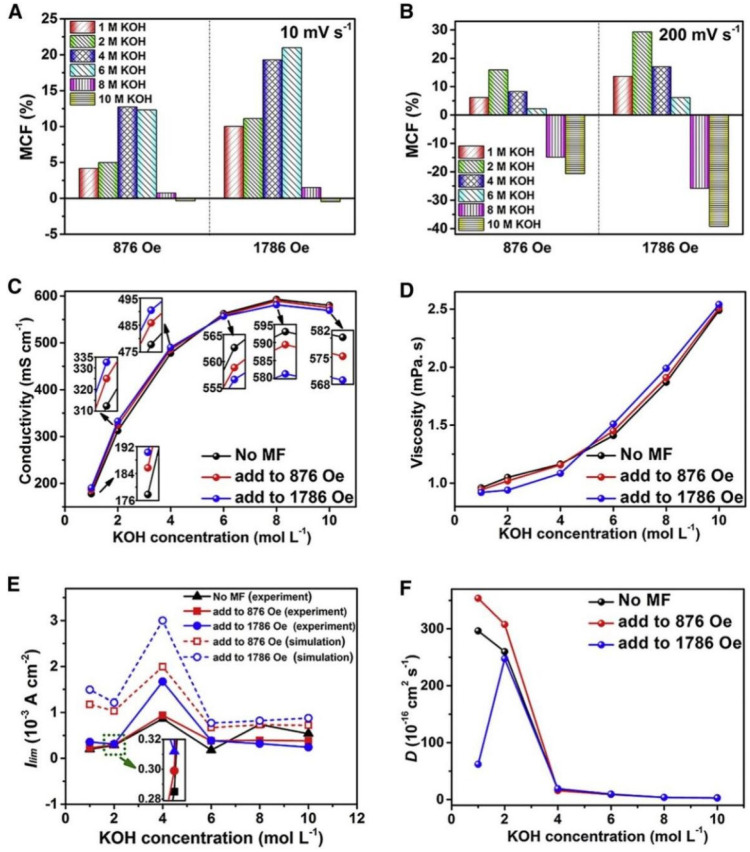
Influence of a magnetic field on the electrochemical properties of KOH electrolytes at various concentrations. Variation in the MCF values at a scan rate of (**A**) 10 mV/s, (**B**) 200 mV/s, (**C**) Variation in the conductivity with KOH concentrations under different magnetic field, (**D**) Variation in the viscosity with KOH concentrations under different magnetic field, (**E**) Variation of I_lim_ with KOH concentrations, and (**F**) Diffusion coefficient of KOH with various concentrations. Reproduced (modified) from [143] Copyright (2021) with permission from Elsevier.

**Figure 17 micromachines-13-01792-f017:**
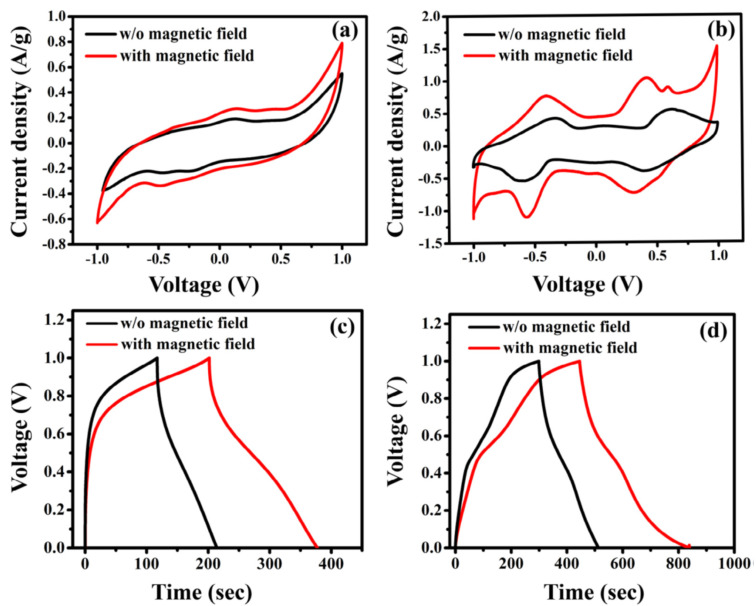
CV and GCD curves of (**a**,**c**) Fe_3_O_4_ and (**b**,**d**) Fe_3_O_4_/RGO with and without magnetic field. Reproduced with permission from [144] Copyright (2018) IOP Publishing.

**Figure 18 micromachines-13-01792-f018:**
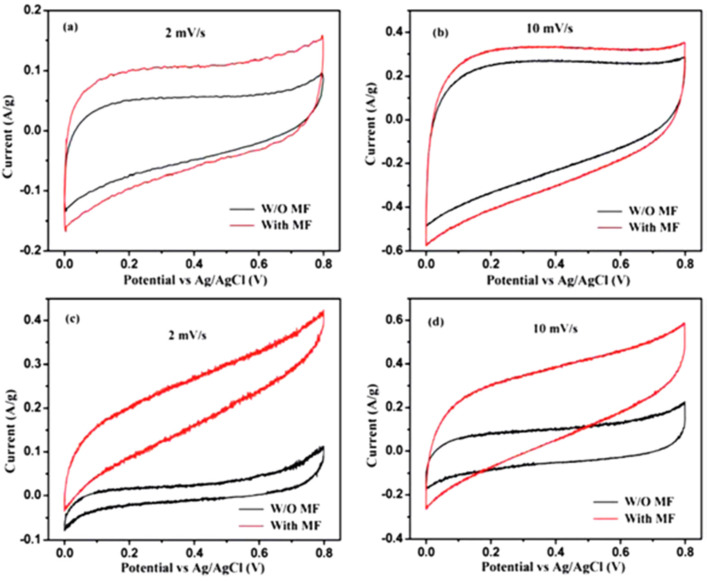
CV curves of (**a**,**b**) Fe_2_O_3_ and (**c**,**d**) Fe_2_O_3_/graphene. Reproduced with permission from [145] Copyright (2012) Royal Society of Chemistry.

**Figure 19 micromachines-13-01792-f019:**
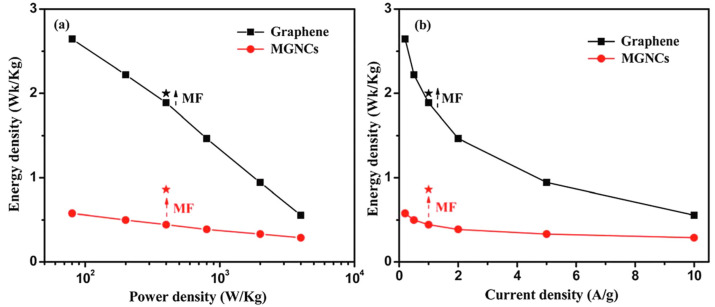
Energy density vs. (**a**) power density and (**b**) current density of graphene and Fe_2_O_3_/graphene composite. Reproduced with permission from [145] Copyright (2012) Royal Society of Chemistry.

**Figure 20 micromachines-13-01792-f020:**
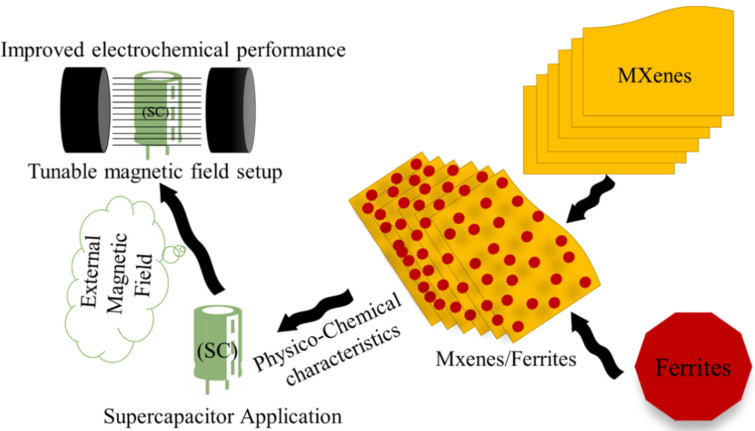
Schematic process diagram for the magnetic field assisted supercapacitor application with MXene/Ferrite magnetic nanocomposites.

**Table 1 micromachines-13-01792-t001:** Saturation magnetization, coercivity and K1 values of spherical, cubical and hexagonal shaped Co-ferrites [63].

Temp	Spherical Co-Ferrite			Cubical Co-Ferrite			Hexagonal Co-Ferrite		
	Ms (emu/g)	Hc (kOe)	K1 (10^6^)	Ms (emu/g)	Hc (kOe)	K1 (10^6^)	Ms (emu/g)	Hc (kOe)	K1 (10^6^)
10	81.11	17.31	3.11	83.21	17.05	2.91	86.11	17.85	3.38
50	79.46	13.62	2.57	81.14	12.75	2.43	84.63	14.09	2.78
100	76.94	10.36	2.31	78.18	9.48	2.11	82.74	11.38	2.44
150	75.08	6.64	1.89	77.24	6.35	1.78	80.73	6.81	2.24
200	73.55	4.47	1.48	74.72	3.89	1.34	78.48	4.67	1.75
250	72.9	3.02	1.21	73.14	2.83	1.23	76.15	3.18	1.42
300	70.68	1.39	0.98	71.74	1.09	0.89	75.32	1.58	1.11

**Table 2 micromachines-13-01792-t002:** MXene composites and their applications.

S. No	Composites	Synthesis Method	Applications	Ref.
1	MXene/NiFe_2_O_4_	In situ chemical co-precipitation method	microwave absorption	[82]
2	MXene/NiFe_2_O_4_	ultrasonication	wastewater treatment and antibacterial activity	[83]
3	MXene/NiFe_2_O_4_	one-step hydrothermal method	microwave absorption	[84]
4	MXene/Ni_0.5_Zn_0.5_Fe_2_O_4_	co-precipitation process	electromagnetic wave absorption	[85]
5	MXene/CoFe_2_O_4_	In situ solvothermal process	microwave absorption	[86]
6	MXene/CoFe_2_O_4_	liquid self-assembly method	catalytic degradation of Naproxen	[87]
7	MXene/CoFe_2_O_4_	hydrothermal synthesis	electromagnetic wave absorbing	[88]
8	MXene/CuFe_2_O_4_	sol-gel/sonication	photocatalytic and antibacterial	[89]
9	MXene/La and Mn-co-doped Bismuth Ferrite	double-solvent sol–gel method	photocatalytic degradation of congo red dye	[90]
10	MXene/Fe_3_O_4_-myristicacid	in situ chemical synthesis strategy/physical vacuum infiltration	photothermal conversion	[91]
11	MXene/CoFe_2_O_4_	sonication	supercapacitor	[24]
12	MXene/Fe_3_O_4_/RGO	modified chemical oxidation process	supercapacitor	[92]
13	MXene/Fe_3_O_4_/MXene	laser crystallization strategy	flexible supercapacitor	[93]
14	MXene/Co_0.7_Zn_0.3_Fe_2_O_4_/MWCNT	solvothermal process	electromagnetic absorption	[23]
15	MXene/Ni_0.6_Zn_0.4_Fe_2_O_4_	In situ hydrothermal method	electromagnetic wave absorption	[94]
16	MXene/NiZn Ferrite	co-precipitation hydrothermal method	electromagnetic devices applications	[95]
17	MXene/Co-doped NiZn ferrite/polyaniline	exfoliation, hydrothermal process, and modified interfacial polymerization	electromagnetic wave absorption	[96]
18	MXene/Fe_3_O_4_/PANI	vacuum-assisted filtration	electromagnetic interference shielding	[97]
19	MXene/Fe_3_O_4_@PANI	co-precipitation and in-situ polymerization	wide-band electromagnetic absorber	[98]
20	MXene/Fe_3_O_4_/Au/PDA	hydrothermal/freeze dry	photo-thermal-magnetolytic coupling antibacterial	[99]
21	MXene/Fe_3_O_4_	hydrothermal reaction	water splitting	[100]
22	MXene/Bismuth Ferrite	double-solvent solvothermal method	degradation of organic dyes and colorless pollutants	[101]
23	MXene/Fe_3_O_4_/C	microwave-assisted reaction, thermal treatment and sonication	electromagnetic absorption	[102]
24	MXene/Ba_3_Co_2_Fe_24_O_41_/Polyvinyl Butyral	tape-casting process	electromagnetic wave absorption	[103]
25	MXene/Fe_3_O_4_/TiO_2_	in situ oxidation	phosphoproteomics research	[104]
26	MXene/Fe_3_O_4_/MWCNTs	solvothermal and vacuum-assisted filtration method	electromagnetic interference shielding	[105]
27	RGO/MXene/Fe_3_O_4_	ultrasonic spray technology	microwave absorbents	[106]
28	Fe_3_O_4_/MXene/polyurethane	--	stretchable electromagnetic shielding	[107]
29	MXene/Fe_3_O_4_/polyimide	In situ polymerization, electrospinning technology and vacuum-assisted filtration	electromagnetic interference shielding, electrothermal and photothermal conversion	[108]
30	MXene/Fe_3_O_4_	thermal treatment	heterogeneous Fenton reaction catalyst towards MB, MO, RB and CR	[109]
31	MXene/Fe_3_O_4_	chemical and filtration	removal of cationic dyes from aqueous media	[110]
32	MXene/Fe_3_O_4_	ultrasonication	Li-ion batteries	[111]
33	MXene/Fe_3_O_4_	hydrothermal and vacuum filtration process	heavy metal ions removal from wastewater	[27]
34	MXene/Fe_3_O_4_	freeze-drying/freeze drying	Li-ion batteries	[112]
35	MXene/Fe_3_O_4_	thermal decomposition process	microwave absorbing	[113]
36	MXene/Fe_3_O_4_	biomimetic strategy-polymerization/sonication	photocatalytic degradation dyes	[114]
37	MXene/Fe_3_O_4_	chemical hydrothermal reaction	microwave absorption	[115]
38	MXene/Fe_3_O_4_	self-assembly approach	lithium-ion storage	[116]
39	MXene/Fe_3_O_4_	solvothermal method	microwave absorbers	[81]
40	MXene/Fe_3_O_4_	etching, sonication, self-assembly	nonlinear optics	[117]
41	MXene/Fe_3_O_4_	hydrothermal reaction method	microwave absorption	[118]
42	MXene/C/Fe_3_O_4_	facile chemical process	Li-ion batteries	[119]
43	MXene/Fe_3_O_4_	solvothermal process	microwave absorption	[120]
44	MXene/Fe_3_O_4_	chemical and sonication	lightweight electromagnetic absorbers	[121]
45	MXene/Fe_3_O_4_	in situ growth method	Li-ion storage	[122]
46	MXene/carbon coated Fe_3_O_4_	--	Li-ion storage	[123]
47	waterborne polyurethane/MXene/NiFe_2_O_4_	freeze-drying method	electromagnetic interference shielding	[124]
48	Fe_3_O_4_/MXene/carbon nanofibers	--	Li-ion batteries	[125]
49	Alk-MXene/Fe_3_O_4_	ultrasonic treatment	lithium−sulfur batteries	[29]
50	MXene/Fe_3_O_4_-COOH	rapid gelation method	human motion monitoring	[126]
51	MXene/Fe_3_O_4_/3,4-dihydroxyphenylacetic acid (DOPAC)-epoxidized natural rubber (ENR) elastomers	--	flexible electromagnetic interference shielding	[127]
52	MXene/NiFe_2_O_4_/Carbon Felt	facile dip-and-dry and hydrothermal methods	microbial fuel cell	[128]
53	MXene/PPy/β_2_-SiW_11_Co/Fe_3_O_4_	etching/sonication	electromagnetic wave absorption	[129]
54	Fe_3_O_4_-MXene-Carbon Nanotube	chemical precipitation method	supercapacitor	[130]
55	rGO/MXene (Nb_2_CT_x_)/Fe_3_O_4_	hydrothermal treatment and self -assembly	absorption of electromagnetic wave	[131]

**Table 3 micromachines-13-01792-t003:** Utilization of external magnetic field for the synthesis and applications.

S. No	Materials	Magnetic Field (T)	Synthesis/Applications	Improved Efficiency (%)	Ref.
1	FeCo_2_O_4_	3 mT	supercapacitor	56	[146]
2	AC/Fe_3_O_4_	--	supercapacitor	33.1	[147]
3	Fe_3_O_4_/PPy	0.1 T	supercapacitor	128	[148]
4	Mn_3_O_4_	Up to 3 mT	supercapacitor	50	[149]
5	Nanoporous nickel	500 mT	supercapacitor	~12	[150]
6	Graphene/Fe_2_O_3_	72 mT	supercapacitor	154	[145]
7	Graphene oxide anchored Fe_3_O_4_	0.125 T	supercapacitor	193	[144]
8	FeCo_2_O_4_	3 mT	supercapacitor	~21	[151]
9	Flexible magnetic Microtubule nanocomposite fabrics ((Iron, Nickel, Cobalt) with carbon)	72 mT	supercapacitor	70	[152]
10	Ferromagnetic metal oxide -Fe_2_O_3_	0 to 5 mT	supercapacitors	170	[153]
11	Co_3_V_2_O_8_-RGO	0.5 T	supercapacitors	∼170	[154]
12	NiCo-graphene quantum dot	--	supercapacitor	-	[155]
13	MnO_2_-electrospun carbon nanofibers	1.34 mT	supercapacitor	19	[156]
14	α-Fe_2_O_3_/N-doped carbon	0.15 T	Li-ion battery	45	[157]
15	KCl, KBr, KI, Na_2_SO_4_, and CH_3_COOH	15 mT	improvement in conductivity	-	[158]
16	Iron	0.6 T	ferromagnetic electrodes	-	[159]
17	Water-electrolyte solution	0.42 T	Ion-Exchange Kinetics	-	[160]
18	Fe_3_O_4_/TiO_2_	Up to 80 mT	photocatalysis	~200	[161]
19	ZnFe_2_O_4_	Up to 0.8 T	photocatalysis	150	[162]
20	α-Fe_2_O_3_-decorated TiO_2_ nanotube arrays	0.4 T	photocatalysis	∼38	[163]
21	Au-CdS	Up to 81 mT	photocatalytic hydrogen production	110	[164]
22	TiO_2_	0.28 T	photodegradation of methyl orange	24	[165]
23	Palladium	0.5 T	catalytic activity in Suzuki cross-coupling reactions	50	[166]
24	Copper metal ions	0.685 T	magneto electrolysis	-	[167]
25	NiZnFe_4_O*_x_*	≤ 450 mT	electrocatalytic water oxidation	100	[168]
26	AgAu	450 mT	electrochemical dealloying	With MF − 0.5 cycles and without MF > 1.5 cycles	[169]
27	NiO on nickel foam	0–12.6 mT	magnetic-field-assisted hydrothermal method	-	[170]
28	Copper	0.4 to 1 T	electrodeposition	-	[171]
29	Silver	1.2 T	electrodeposition	-	[172]
30	Nickel	1 T	electrodeposition	-	[173]
31	Cobalt	0–5 T	electrodeposition	-	[174]
32	Nickel	0.7 T	electrodeposition	-	[175]
33	Cobalt hydroxides	2 T	phase engineering	-	[176]

## Data Availability

Not applicable.

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
