# Peer review of "MXene/Ferrite Magnetic Nanocomposites for Electrochemical Supercapacitor Applications"

_micromachines, 2022, doi:10.3390/mi13101792_

Round 1

Reviewer 1 Report

The review paper "MXene/Ferrite Magnetic Nanocomposites for Electrochemical Supercapacitor Applications" is devoted to the combination of 2D MXene material with ferrite nanoparticles and the use of these composites in the application of supercapacitors. The paper is well organized. There are parts about 2D MXene and ferrites synthesis, their composite formation, and electrode formation for supercapacitor applications. The authors draw attention to the use of a magnetic field to change the parameters during an electrochemical reaction, which can lead to an increase in the capacitance of a supercapacitor, based on the literature data, in some cases by a factor of two. 

I think the review paper can be published in Micromachines.

Author Response

Thanks for the positive comments. We have improved the manuscript by checking for English corrections and a few figures were replaced with high-quality figures.

Reviewer 2 Report

Thirumurugan et al. summarized the application progress of MXene/ferrite magnetic nanocomposites for electrochemical supercapacitor. Based on my understanding of this field, this is a good and interesting review article. My overall response to this work is positive, and encourage the publication of this work with minor revisions:

1. MXene is easy to be oxidized and decomposed under the combined action of water and oxygen, which leads to low quality and short storage period of MXene dispersion, which is also a huge challenge affecting the transition of MXene from laboratory to industrial application. Therefore, the author should provide detailed insights on the preparation of high-quality MXene in the prospect section to improve the quality of contributions.

2. The resolution of some pictures in the manuscript has declined. It is recommended to replace high-quality pictures.

3. The cohesion and language between some sentences in the manuscript need to be improved, otherwise the readability of the article will be affected.

Author Response

Thanks you for the positive comments. 

We have improved the manuscript as per the reviewer's suggestions.

Response to Reviewer 2 Comments

Point 1. MXene is easy to be oxidized and decomposed under the combined action of water and oxygen, which leads to low quality and short storage period of MXene dispersion, which is also a huge challenge affecting the transition of MXene from laboratory to industrial application. Therefore, the author should provide detailed insights on the preparation of high-quality MXene in the prospect section to improve the quality of contributions.

Response 1: Yes. MXene storage is important for any application as it is changing its characteristics based on environmental characteristics. The importance of MXene storage in particular ways to improve the MXene characteristics by avoiding oxidation and decomposition is discussed in the revised manuscript. 

The following sentences has been included in the revised manuscript:

“The stability of MXene is one of the important parameter to be considered for any kind of application [48,49]. In general, the MXene easily gets oxidized or degraded from its original physicochemical characteristics. The oxidation of Ti3C2 resulted in Ti-based oxide materials on the surface or anywhere in the MXene along with amorphous carbon. However, several procedures were demonstrated to control the oxidation or degradation process of MXenes[50]. A few of the process includes the synthesis of MXene with fewer defects, surface modification of MXene with organic ligands or any other suitable surface modifier, dispersions of MXenes in organic solutions, and the storing of MXene in solution which is stored inside the inert gas-filled container[49,51]. If the MXene hybrids with polymeric materials or any other stable materials are required in particular applications then the degradation of stability and electronic conductivity of MXene could be reduced due to the polymeric or any other stable materials in hybrids[52]. ”

Point 2. The resolution of some pictures in the manuscript has declined. It is recommended to replace high-quality pictures.

Response 2: The high-quality figures were replaced

Point 3. The cohesion and language between some sentences in the manuscript need to be improved, otherwise the readability of the article will be affected.

Response 3: The language correction is done in the revised manuscript. Some of the sentences have been modified in the revised manuscript.
